# An Investigation of the Use of Augmented Reality in Public Art

**Tamlyn Young** [1,2] **and Mark T. Marshall** [2,*]

1    Limerick School of Art & Design, Technological University of the Shannon, V94 KX22 Limerick, Ireland; tamlyn.young@tus.ie
2    Interaction Design Centre, University of Limerick, V94 T9PX Limerick, Ireland
\*    Correspondence: mark.marshall@ul.ie

**Abstract:** Augmented reality offers many artistic possibilities when it comes to the creation of place-based public artworks. In this paper, we present a series of works around the topic of augmented reality (AR) art and place-based storytelling, including the use of walking as a creative method, a series of workshops with emerging artists, public AR art collaborations and a study to examine user experience when interacting with such artworks. Our findings from these works show the potential of integrating augmented reality with public physical artworks and offer guidance to artists and AR developers on how to expand this potential. For artists, we show the importance of the space in which the artwork will be placed and provide guidance on how to work with the space. For developers, we find that there is a need to create tools that work with artists' existing practices and to investigate how to expand augmented reality past the limitations of site- or piece-specific apps.

**Keywords:** augmented reality; public art; visitor experience; place-based storytelling; location-based applications

## 1. Introduction

In this paper, we explore the potential of artist-oriented augmented reality (AR) to expand traditional image-making practices. Augmented reality art practice is currently evolving as the availability of AR tools and platforms increases, and as a result, "there is a growing need to question and re-evaluate its potential as a medium for creative expression" [1].

By taking these practices beyond the studio and the gallery and into an urban environment, we aim to evaluate the capacity of AR Art to influence public engagement with overlooked spaces. Through this work, we interrogate the role of graphic art in the socio-cultural structure of a community and contribute to the current discourse around the role of AR in art practice. Our work aims to promote collaboration between emerging artists and designers in the community and to transform "invisible" spaces into storied spaces by inviting members of the community to engage with the urban environment in new ways.

A component of the research is to work within the affordances and limitations of leading artist-oriented applications. Applications such as Artivive and Adobe Aero are designed to be accessible and user-friendly, with a minimal to moderate learning curve for artists and designers without a background in computer programming. These applications place AR Technology within the reach of creatives, thereby expanding its potential applications beyond traditional scientific and commercial uses. This is a move toward the democratisation of content creation, allowing for imaginative and innovative multimedia storytelling. For creatives, this poses the question, "Now that we have all of this tremendous technology, how are we going to use it?" [2].

Our interest here is in the potential of AR to challenge conventional approaches to drawing by facilitating the intersection of physical with digital space to create immersive, multi-layered visual narratives. While there is an increasing number of artists and designers

practicing at this intersection, there remains a paucity of practice-based academic research that explores and consolidates this emerging area of interest. There is also a lack of knowledge about the usability of interactive AR in such a context and the technological and interaction requirements that would enable artists and designers to create such narratives and enable users to understand and interact with them.

In contemporary society, our trajectories and interactions are increasingly mediated by technology. It is argued that the technologies we are creating have the effect of alienating or subverting our grounding in physical space, affecting our sense of community and active participation in the public realm [3]. By contrast, AR is described as an inhabited environment in which the physical and the digital "co-produce and co-construct one another" [1]. This capacity of AR to add a digital layer that interacts with, rather than obscures, our situated context has the potential to re-establish the user's connection to physical space, helping one be more present and engaged while stretching social and perceptual boundaries [4]. Indeed, it has been noted that "AR affects not only the space that is augmented but also the users/creators of the technology and how they think about space" [5]. To ensure that AR is used to enhance rather than colonise physical space, it is important to create disciplines and cultivate practices that are inclusive and positive [3]. This observation suggests that how this emerging technology evolves and impacts society depends on who has access to creating with it and what their intentions are.

Our research questions focus on how artists and designers can integrate AR into their work when creating place-based stories, how AR can add additional narrative layers to public artworks, and the benefits to both artists and audiences from the use of AR in such works. In this paper, we discuss our process and practice in the creation of AR artworks in public spaces, present a series of works that have taken place as part of this research and discuss what we have learned from these works and how these learnings can be of benefit to researchers and practitioners interested in augmented reality and its use in interactive public artworks.

The structure of this paper is as follows: Section 2 discusses existing work on augmented reality artwork and place-based storytelling. Section 3 introduces technologies for augmenting public spaces. Section 4 discusses the research methods we have used as part of the practice-based research methodology we are following in this project. Section 5 presents an exhibition of augmented reality artworks that was run in order to gain feedback from the public on the benefits and issues around placing such artworks in public spaces. Section 6 discusses our findings from the projects and offers suggestions for artists and technologists interested in AR artwork. Finally, Section 7 presents our conclusions and future directions.

## 2. Augmented Reality Art and Place-Based Storytelling

Augmented reality allows for the creation of artworks that combine both the physical and the digital, enabling artists to layer digital content on top of physical spaces and artworks. In doing so, it allows art to be experienced in places where art is not normally located and where the audience may not have directly sought it out but rather encountered it as part of their everyday experience [6].

It is this unexpected, out-of-place nature of augmented reality art that gives it its specific impact, allowing it to become "art that operates by ambush, rather than asking you to pay up before you see it" [7]. Indeed, this novel interaction of technology, art and place could be seen to separate augmented reality art from other forms, becoming not "art", but "ARt" [8].

While AR art has grown in popularity, with a large number of artists working with this technology both for standalone and place-based artworks (for examples, see [9–12]), there has been little academic work on how AR can be incorporated into artistic practice (with some exceptions, such as [13,14]), meaning that for most artists working with AR, they have to learn from scratch, rather than being able to build on the shoulders of those who have worked with this technology before. This is not to say that there are no works

available that discuss the creation of augmented reality artworks, but rather that these focus on the creation of those specific works rather than reflecting on how an AR art practice can be developed.

When combined with place-based storytelling, augmented reality allows for a new form of digital placemaking, where the technology is used to tell the story (or stories) of the place or of those people who live in that place [15]. There are many places worldwide with stories attached to them—the sites of major historical events; historical buildings and infrastructures; and places where historical figures were born, lived or died. Each of these stories can be (and are) told in different ways through history books, guided tours and commemorative plaques. However, augmented reality allows us to tell these stories in new ways, to overlay digital content onto the physical space, potentially combining photos, videos, audio, animations, illustrations and more with the physical reality of the place. For example, the Museum of London's StreetMuseum app overlays digital content (such as historical photos) over the physical place where that photo was taken, allowing the user to view the photo in the context of its place in the modern world and examine the differences between then and now [16].

The combination of AR and place-based storytelling also allows for the presentation of a range of stories. While physical artefacts such as commemorative plaques can only commemorate one aspect of a place, AR can be used to present multiple different stories in the same place. This could range from telling stories collected from current residents [17,18] to fictional stories where the places form part of the underlying plot [19].

## 3. Augmenting Public Spaces

There are a number of available technologies in order to augment reality with digital content. These can be mainly broken down into three categories: marker-based, markerless and location-based. Marker-based AR makes use of 2-dimensional marker images that are easily recognised by the image-tracking capabilities of a range of augmented reality frameworks [18]. In general, it is considered that marker-based AR is best suited to applications involving a fixed environment or where the entirety of the virtual content can be viewed while keeping the marker in view of the camera [20]. Marker-based tracking has also been extended to employ 3D markers for improved tracking [21], including calculating properties such as translations and rotations.

Markerless AR, on the other hand, uses computer vision techniques to detect features in the environment (such as flat surfaces, for example) and overlay the augmented reality content on these features. Once the content has been placed on a feature, the user can move around the content and view it from different angles and distances with more freedom than allowed by marker-based augmented reality [18,20].

Finally, location-based augmented reality makes use of sensors within the viewing device to determine physical location and orientation. By combining GPS and compass data, it is possible to determine both where the users' device is located in space and what they are looking at. Content can then be placed at fixed locations in real-world space and viewed from a variety of distances and angles by users within that space. However, it should be noted that the actual placement of content in the real-world space can be somewhat "fuzzy" in that exact positioning can be difficult depending on GPS coverage, with issues of signal interference and overshadowing being particularly prevalent in dense city areas [22].

In our work presented here, we made use of a form of marker-based augmented reality. The markers that we used are large illustrations and/or murals that are placed in the locations where the augmented reality content is to be displayed. These act as visual markers, but their large size allows for a much larger interaction space while making them easier to detect than traditional AR markers. This form of augmented reality marker has proven quite successful in the augmenting of urban spaces, such as those in the Bowery Wall and Parabola projects [15]. We also chose this method as it is among the

best-supported tracking methods in the AR tools aimed at artists and so fits best with the scope of our research.

## 4. Using Augmented Reality with Public Artworks

The works described in this paper take place within the context of practice-based research into the use of augmented reality for place-based storytelling. We are particularly interested in how place-based storytelling with augmented reality art can contribute to revitalising public spaces. As part of this, we investigate issues around both the use and creation of augmented reality art for public spaces.

From the point of view of use and interaction, we focus on how the user experience can be designed in a manner that is accessible and adds value both to viewers' experience of the artwork and the place in which it is installed. One of the challenges of putting AR art in public spaces is that the public does not necessarily know that it is there, so how can we design the real-world aspects of the artworks in order to engage potential users and inform them of the presence of the AR aspect? Once they are aware that the AR is present, how do they engage with it?

Having actually enabled interaction between the user and the AR artwork, we then have to consider how the AR art can be designed in order to "occasion a dialogue" with the place and the physical artwork in that place [23]. The AR experience must be designed around the story, centred on the message rather than on the medium. It must be underpinned with a compelling narrative. Otherwise, it is only a gimmick that will not receive a second look.

When designing the AR experience, we believe it is important to consider the relationship between the AR artwork, the architecture of the site and the physical artwork that serves as a marker to trigger the AR. Each of these three elements works together to communicate the story. This relationship must be synergistic and balanced. For example, too much digital content creates a virtual overlay that conceals the physical site rather than revealing new layers of story. In comparison, too little digital content can make the AR seem like a gimmick, something that does not add to the artwork or story in a meaningful way. The concept of place-based storytelling means that the location must also play an important role in the artwork—the artwork must be situated in this place for a reason that is connected to the story.

From the point of view of the creation of AR place-based stories, we must also consider the artists who would be creating the artwork and telling the stories. Are emerging artists interested and willing to engage with AR technology as an output and extension of their work? Some of the projects discussed in the next section of this paper deal specifically with this issue, looking at emerging artists and how we can help them incorporate AR-enabled place-based storytelling into their work.

To enable artists to work with augmented reality in this way, we need to think about how to design for these experiences. What are the processes, workflows and procedures that need to be navigated to connect personal practice in a social space using AR art? What are the perceived affordances and limitations of the technology? Augmented reality artworks exist simultaneously in the physical and the digital worlds. Many existing art practices focus on one or the other, so combining both the physical and the digital may require new tools, new techniques, new workflows. We examine this further in the next section.

While these aspects cover the process and means of creating AR artwork, for artists, the main import lies in the artwork itself. Some have argued that augmented reality should be used to enhance and extend physical artworks rather than turning them into something completely different [24]. Indeed, much of the work we are discussing in this paper takes this approach, focusing on the "augmented" part of augmented reality to extend physical artworks. However, in most cases, the physical artworks have also been specifically created to work with the AR.

Given this combination of physical and digital artworks, the artistic process must incorporate both. The two parts of the artwork should each tell part of the story, so they

must be created to work together. However, as there is no guarantee that viewers of the physical artwork will engage with the AR artwork, should the physical artwork stand alone? Should it tell part or all of the story, or should the storytelling only occur through the augmented reality artwork?

It is also possible for augmented reality to tell multiple stories or tell stories from multiple perspectives. As already mentioned, places can have a number of stories associated with them, so augmented reality can potentially be used to layer these different stories or perspectives on top of the physical artwork. An interesting example of this is the Bowery Wall project already mentioned, which allowed people to view not just the current mural physically painted on the wall but to use AR to overlay previous murals that had been on that wall [15]. Telling multiple stories or showing multiple perspectives may also encourage viewers to engage more than once with the AR experience, something that may not happen otherwise.

## 5. Methods

In this section, we detail several ongoing projects investigating different aspects of place-based storytelling and augmented reality. These projects include workshops with emerging artists, walks, public art collaborations and an exhibition of AR art pieces. Each of these projects forms part of our overall practice-based methodology.

### 5.1. Workshops

To better understand how emerging artists can make use of augmented reality and use it to tell stories with their artwork, we ran three workshops, two in Ireland and one in Portugal.

The first of these workshops took place at Limerick School of Art and Design, Ireland. The 30 participants were undergraduate students in graphic design and printmaking. This workshop involved training in how to use augmented reality to extend and add narrative dimension to a 2D, still piece of visual communication. This included an emphasis on how the concept/message can be supported and strengthened through the introduction of an AR component and the synergy between the still/printed image and AR component in terms of visual communication.

Students produced a total of 20 augmented reality-enabled posters on the theme of "Women's Rights are Human Rights", which were later publicly displayed. Figure 1 shows one of the students working on their poster alongside the resulting (AR) poster. On completion of the project, a web-based survey was circulated amongst the participants. Feedback from the students showed that they perceive AR as an exciting medium with the scope to extend their practice in their respective creative fields, and many said they would employ the technology in their future projects—although mostly as an alternative or supplementary output. This would seem to indicate that for many of these students, the traditional visual media are still considered the most important.

The public display of the outputs of these workshops took place in a gallery space of the institution the students attend. The pieces were on display for one week, and during this period, the augmented reality components received over 3000 views. Viewing the AR required them to download and install an app on their phones, which is, in many ways, a barrier to the experience. The fact that many people did this and that there were many views of the artworks in this period would suggest that visitors were interested in the AR experience. This issue of the app acting as a barrier to interaction with the pieces is common in AR and was also brought up by the students in the workshop as a potential issue. Lastly, while this shows a good level of interest in AR in a gallery space, it does not predict how well this would translate to an outdoor public experience.

The second workshop took place in Barcelos, Portugal. This was an interdisciplinary workshop with 11 international university students arranged as part of the RUN EU initiative (https://run-eu.eu/, assessed on 1 August 2023). This workshop looked at drawing and technology as a means to reflect on the notion of cultural heritage and

document individual responses to the architecture and artifacts produced as part of a cultural heritage. As part of this, students were taught how to bring their sketches of the city and its cultural heritage to life using augmented reality by adding an extra layer of narrative using digital content.

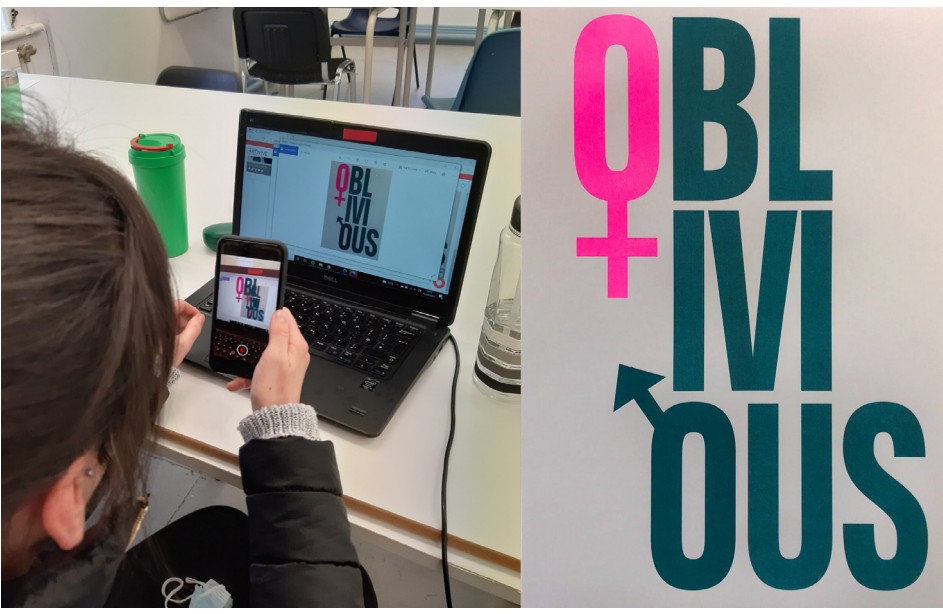

**Figure 1.** (**left**) Workshop participant working on AR poster. (**right**) Final poster. When viewed through the Artivive (https://artivive.com/, assessed on 1 August 2023) app, the AR content is activated. The AR content can also be seen in Video S1.

Prior to the workshop, students had created sketchbooks of the city and its cultural heritage. These sketchbooks acted as a repository of images and stories from the city, with individual sketches acting as marker images for augmented reality. Figure 2 shows some of the sketchbook images generated by the students. This use of a book of marker images raises some interesting questions. Does the artwork need to be installed in the place to be relevant, or could it be easily accessed in a book format? Could the physical, site-specific installation be complimented with a book that also has the AR experience connected to the images? Could the book serve to contextualise the project and incentivise people to visit and engage with the actual sites [25]?

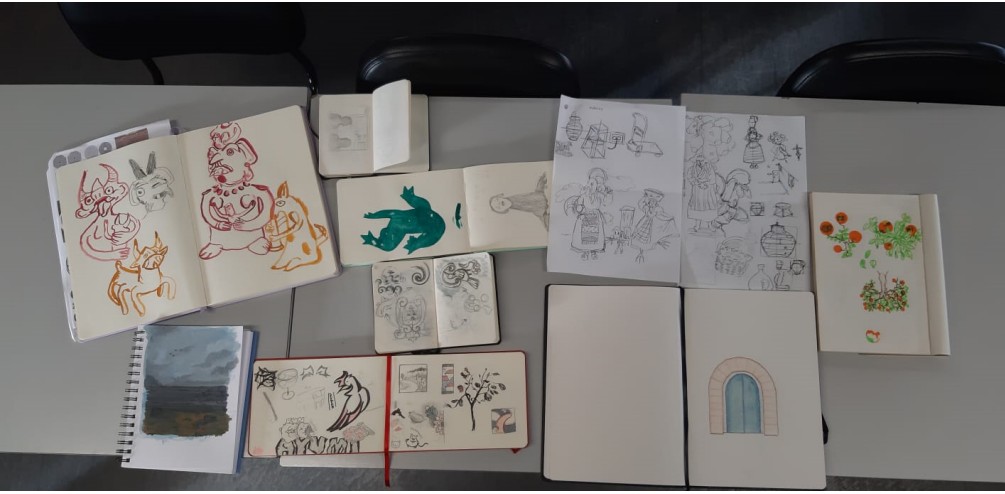

**Figure 2.** A selection of sketchbooks from the students. Some of the final AR artworks can be seen in Video S2.

The third workshop looked specifically at developing illustrations and AR augmentations for use in a public space. This again took place in Limerick, Ireland, and was a collaboration between undergraduate art students and stakeholders in a city-centre regeneration project called the Opera Centre development. The goal of this workshop was for the students to develop AR illustrations that tell the story of what is happening behind the barriers of a regeneration project in a historical part of the city. Given that while the regeneration work is happening, passers-by cannot see the site as it has been surrounded by temporary walls, can we use AR illustrations mounted on these walls to tell the story of what is happening there and why? Figure 3 shows a user interacting with one of the murals.

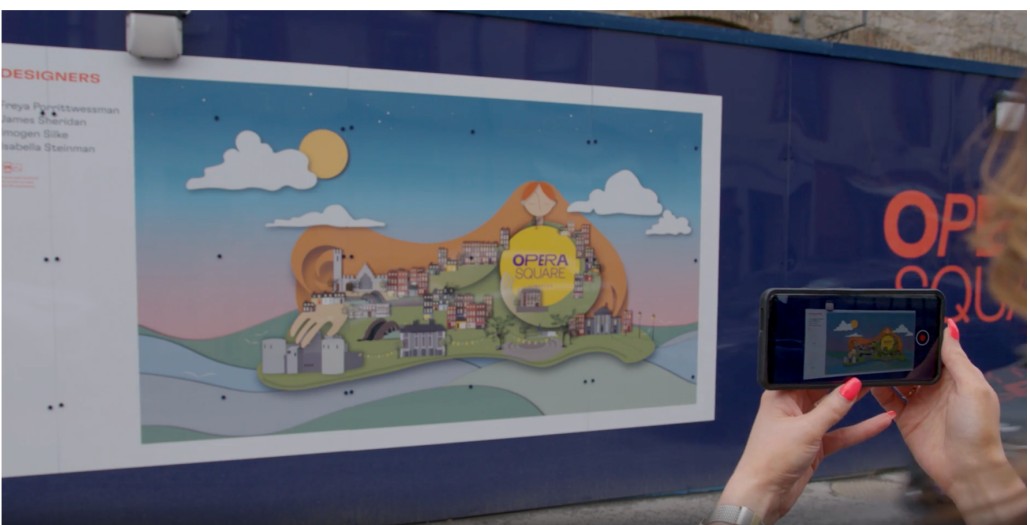

**Figure 3.** A user interacting with one of the murals at the Opera Centre development.

The students were trained not just on the creation of the AR layer for their illustrations but also guided through how to augment their illustrations to extend the messaging of their designs to inform and edutain: to create an interactive experience for members of the public walking past the site. This workshop brought to light some issues with such projects. For example, students sometimes had stories about the site that they wished to tell but that did not fit with the values the other stakeholders wanted to portray. As discussed earlier, sites often have multiple stories attached to them, and AR allows us to potentially tell many of them, but when working in public spaces, we are sometimes limited by the views of people other than the artists.

This workshop also required thought on how to make the AR component accessible in a public space. As with the other works discussed in this section, the AR required the installation of an app. Signposting had to be added to the site to explain this to the public so that they could install the app and interact with the artwork. Again, this can be a common problem with augmented reality artworks; how can we let people know the AR is there and how to access it?

### 5.2. Walks

Walking has been used as a form of ethnographic research when researching urban spaces, a "mobile and embodied practice" that can "offer insights to the multiple splices of time-space narratives" [26]. It has also been acknowledged as a useful tool within art practice, where its multisensory, embodied and emplaced nature allows us to reflect on and engage with the physical world [27].

In this work, walking was used as part of both ethnographic research and arts practice, including individual psychogeography walks, group walking tours and a walk to temporarily install an AR art piece at different locations.

Solo walking following the psychogeographical idea of dérive involves allowing your-self to be "drawn by the attractions of the terrain" [28] and has allowed for an exploration of the urban space, with a particular focus on overlooked/derelict/hidden parts of the city. Different parts of the city elicited different psychological responses: threatening, wel-coming, nostalgic, hopeful. Spaces in which local residents have implemented techniques of tactical urbanism or used creative placemaking to beautify a derelict spot elicit more positive feelings, in direct contrast to the spaces that have been allowed to decay and which seem to have encouraged further damage.

In particular, these walks helped discover the large number of buildings that are abandoned, their stories forgotten or overlooked, such as those shown in Figure 4. Could these spaces be re-invigorated by telling these stories? Could the addition of public AR art help with this by encouraging people to think about the place and its stories, to engage with it and each other and aid in turning it from a decrepit space to one worth saving?

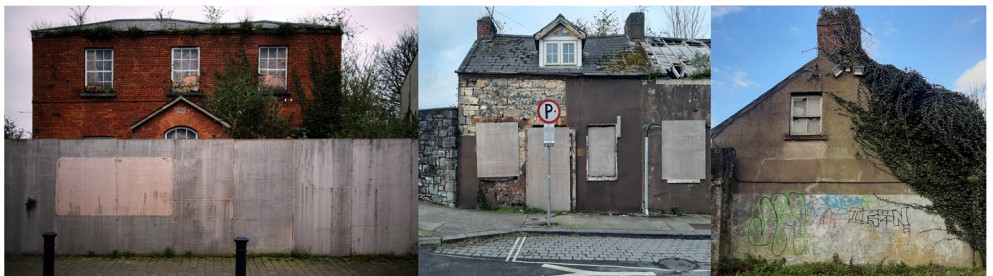

**Figure 4.** A number of derelict sites discovered while walking in Limerick, Ireland.

A second aspect of the walking part of this work was to participate in a group walking tour of the city of Limerick, Ireland. As part of a campaign to highlight vacant and derelict housing in Limerick, over 100 people attended the derelict walking tour of Limerick's city centre on 26 February 2022. These included members of the local public as well as representatives from the city council. A route was plotted to include several key vacant and derelict sites. At each site, a different speaker gave a short talk addressing topics such as the history of the sites, their possible uses and policy solutions to tackle the problem of vacancy.

While this is a guided tour, these organised talks catalysed informal discussions and stories being told by the attendees. People voiced political and personal stories connected to the sites. There is public dissatisfaction with the current state of the sites, with people wondering why no action is being taken with them. One particular site has a history deeply embedded with local lore, yet due to its potential value, it is caught up in political and legal fighting. Such sites are often left to decay while this fighting takes place. Could we instead use art (and AR art in particular) to bring some life back to the sites, to tell the stories and encourage local engagement with them, so they are not completely wasted while waiting on legal decisions?

The final walking project involved walking along an urban canal to paste up a piece of public artwork. Passers-by responded to this process openly, stopping to talk about the artwork and to share their stories about the site. The process of working in the space invites people to pause and chat and share and reflect. This dialogue at least temporarily transforms a liminal space into a place.

The overall purposes of all these walking activities were to investigate the space of the city, find interesting sites, obtain an impression of the stories of these sites and see whether the city and its residents would have stories that could be used to form the basis of a series of augmented reality art pieces.

### 5.3. Public Art Collaborations

Alongside the workshops and walks have also been collaborations on public artworks that make use of augmented reality. In this section, we discuss two in particular, as these were instrumental in influencing the direction of the practice described in this paper.

The first collaboration took place as part of the Waterford Walls project and festival in Waterford, Ireland, in June 2021 (https://wallsproject.ie/waterford-walls-festival/, assessed on 1 August 2023). The Waterford Walls festival is an annual event where national and international artists create large-scale mural artworks around Waterford City and the surrounding areas. The festival includes live art, music, workshops and guided tours. This collaboration involved the creation of an augmented reality artwork to layer over the top of an existing mural, as shown in Figure 5. The goal was to refresh the mural, to bring attention back to a mural that may have become functionally "invisible" due to having been in place for so long. One of the key aspects of this project was to collaborate with the mural artist to ensure that the AR piece acts as an extension of their original concept and does not contradict the original message. If it fails to build on the existing narrative, the AR layer can be perceived as an unnecessary gimmick, an intrusion rather than an addition to the public space. It was this collaboration that showed the power of augmenting public artworks with AR content, with the original mural acting as both an artwork in its own right and as a reliable marker for augmented reality, while the AR content brought an additional layer to the story.

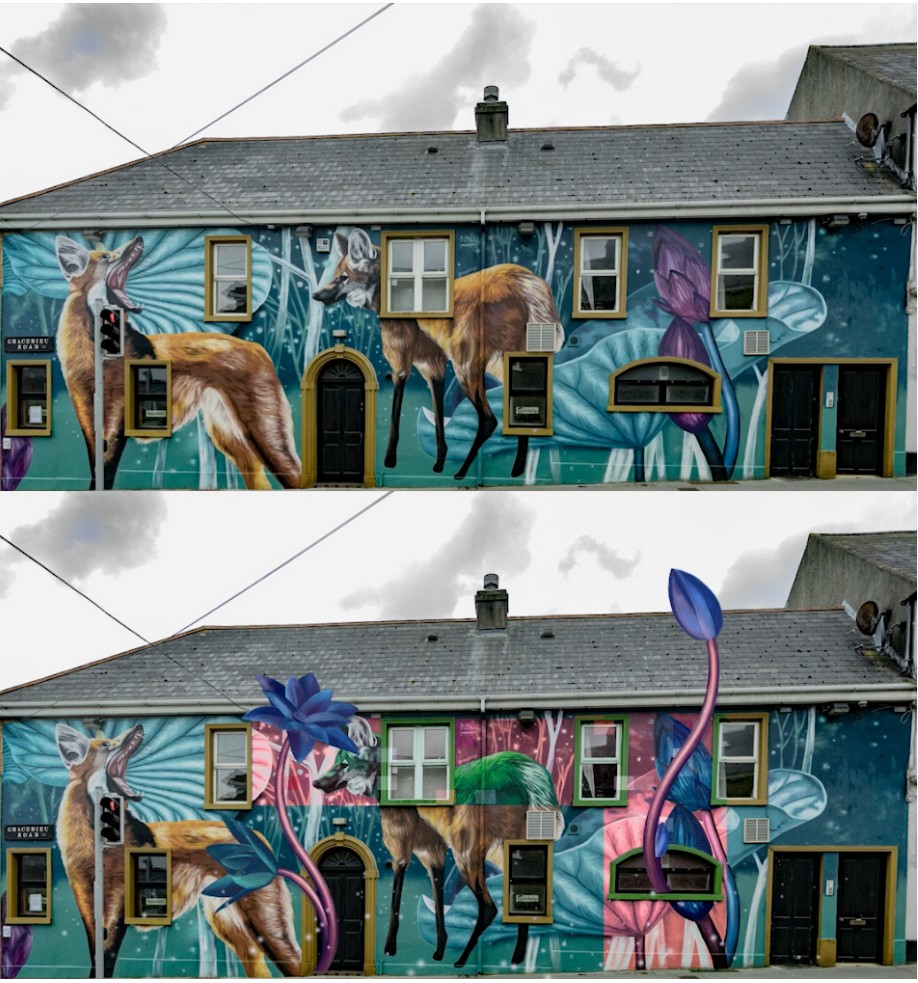

**Figure 5.** (**top**) The mural for the Waterford Walls festival work (**bottom**) A still image from the augmented reality layer on top of the mural. The complete AR content can be seen in Video S3.

The second collaboration took place in August 2021 as part of Birr Vintage Week and Art Festival (http://www.birrvintageweek.com/), an annual community arts festival in Birr, Ireland. This collaboration involved using the architecture of derelict sites around Birr as the basis for a series of AR artworks. The goal of the AR was to be site-specific, interact with the existing architecture and pay homage to it rather than hiding it with a layer of digital content. For this work, a series of physical artworks were created and placed on these derelict spaces and then used to trigger the augmented reality content, such as that shown in Figure 6. These three elements together tell the story: the architecture and story of the physical site, the still image and the AR experience. When designing such an AR experience, it is essential to consider the importance of the relationship between the AR artwork and the existing architecture and the still image that serves as a marker to trigger the AR experience.

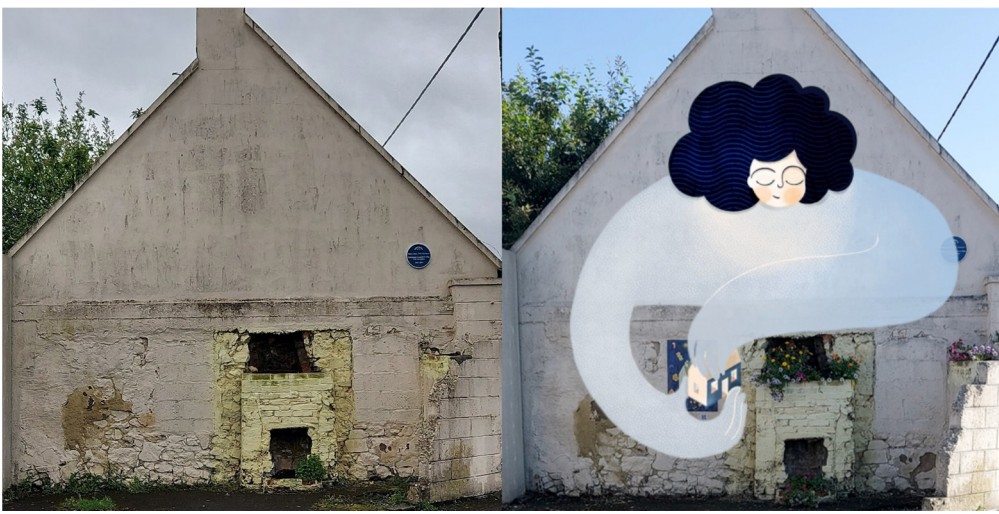

**Figure 6.** (**left**) An abandoned property in Birr, Ireland. (**right**) The augmented reality content overlaid on this property.

Feedback on these collaborations was collected qualitatively through correspondence with sponsors, organisers and fellow artist collaborators. This revealed that the projects had been well received by the communities in which they had been installed, suggesting that there would be scope for further collaborations of this kind.

When taken together, these workshops, walks and collaborations provided inspiration to develop a practice for using physical artworks in urban spaces as triggers for augmented reality content while also fostering an interest in derelict spaces and how these could be brought back to life using these artworks. In the next section, we discuss "Re:Imagined", a series of such artworks created for derelict sites across the city of Limerick, Ireland, as well as an exhibition where the pieces were publicly displayed out of context to examine how well the AR worked and whether people would engage with it.

## 6. Examining Public Response to Augmented Reality Art in the Re:Imagined Exhibition

Re:Imagined was a series of prints and AR content designed as prototypes, the first stage of research into place-based storytelling using augmented reality art. The project aimed to produce a series of "living" illustrations to be installed and experienced in derelict and overlooked sites in the city of Limerick, Ireland. Each illustration was inspired by a (hi)story or memory connected to the site. By embedding stories about a place at the site itself, members of the public were invited to engage overlooked spaces with new awareness. This intersection of art and technology provides a platform for dialogue around how these sites have been and could be used and around notions of home and sense of place, while the use of derelict spaces increases the unexpected aspect of finding art in such a place [6].

To gain some feedback on the AR content and its relationship with the physical artworks, we arranged two exhibitions of prototypes of the works. For these exhibitions, prototypes were created of each of the 10 artworks. High-resolution photographs of the buildings were edited to include the physical artwork and then printed on large format paper. These prints were augmented with digital content, an example of which can be seen in Video S4.

Our research questions for these exhibitions focused on the use of augmented reality content in public artworks. We looked to determine whether passers-by would be aware of the AR artworks and whether they would be willing to engage with them. We were also interested in determining what the barriers were to engaging with the artworks and how this could be improved. Lastly, we aimed to discover whether the experience of the artworks was in any way improved by the addition of the augmented reality layer.

These prints then allowed us to gather feedback from a larger range of potential users by exhibiting them together in public spaces where large numbers of people would pass by. The exhibitions took place in the foyer of the Computer Science Building at the University of Limerick, Ireland, from 27 October 2022 to 7 November 2022 and in the Limerick School of Art and Design from 17 April 2023 to 2 May 2023.

During these periods, 629 people interacted with the artworks. Alongside this, 21 people completed a survey designed to gather their reactions to the artworks, as well as their opinions on how the combination of place, physical art and AR art worked together.

Feedback was generally positive. All of the participants expressed a positive opinion of the use of AR art in public spaces to enliven the cultural landscape of the city. However, it was noted that there was a lack of awareness of the existence of AR art, with 43% (9/21) indicating that if they had seen the artwork in public, they would only have looked at the physical art, as they would have been unaware of the AR content. Comments such as "It would firstly be hard to understand if it's AR art", or asking for some way of "letting me know it's an AR artwork", or making clear "the fact that it was an AR piece", indicated that people did not know they could see an extra level of content. While there were instructional posters placed next to the artworks, it seemed many ignored them and only viewed the AR content when informed of its existence by a facilitator or from watching someone else view it.

Some participants (33%, 7/21) suggested larger, clearer instructions; a large QR code; or some other agreed symbol for AR as a means of informing people that there is digital content available. This would likely be even more of an issue when the artworks are installed in situ, as passers-by do not expect AR artworks in the city and are also less likely to stop and read any instructional signage. This is a common issue with the use of new or novel technologies like AR and has been noted even in places such as museums and galleries where people are more likely to take time to engage [29].

Another common issue noted was the use of a phone app to view the content. Eight participants (38%) either said that the requirement to install an app was cumbersome, expressed a preference of being able to just use their camera app, or would have preferred the augmentations to not require a phone at all. Currently, smartphones are the best available platform for augmented reality, but this may change as AR headsets become more common. However, the idea of being able to use the phone's camera app is an interesting one, as it could result in someone stopping to take a photo of an interesting piece of art and experiencing the AR content without knowing it was there, providing perhaps a more "magical" experience than that already available with current AR apps.

Participants were also asked about how they would interact with the artwork when it was in situ in the city. While most (17/21, 81%) agreed that interacting with the artworks in place would expand their connection with the space itself, several participants raised issues around the ability to really view the artwork in the city—due to foot and vehicle traffic, noise, lack of space, or even lack of time. Indeed, people walking within a city often do so as part of their work, while shopping or running errands, and so it may be hard for them to find the time to stop and interact with the artworks or even notice them.

Our observations of visitors to these exhibitions would, however, suggest that passers-by are more likely to stop and interact with the art if they see someone else doing so, so there is perhaps a potential for this within the city itself. Once enough people are aware of and visiting the artwork, this may cause a knock-on effect of others seeing them interact and being interested in finding out about it.

In terms of reaction to the artworks themselves, participants were positive about the AR aspects. In particular, the AR content was referred to as having "come to life" (8/21 participants, 28%). Positive comments included, "It gives a different dimension and is way more expressive compared to just a 2D artwork" (P8), "It makes it more personal and interactive which makes it memorable" (P16), "I loved the sound it really emersed you and the individuality of each piece" (P18), "I think AR adds extra depth to artwork and makes it more engaging" (P20) and "its mystery is revealed through AR" (P21). One participant noted that they spent longer looking at each image than they normally would have if it were only a series of static images. If we consider the goal of the AR content to have the users engage with the work and spend time in the space, then it seems clear that there is great potential in the use of AR content layers in this way.

Finally, we asked participants the following two questions:

Q1. How likely would you be to view the artwork more than once?
Q2. Having viewed the AR installation, how likely would you be to record, share and recommend the experience?

Their responses were collected on a 5-point scale from "Very Unlikely" to "Very Likely". For Q1, the most common answer was that the participants were "Somewhat Likely" to view the artwork more than once, while for Q2, the most common answer was that participants were "Very Likely" to record, share and recommend the experience. Figure 7 shows the participants' responses to these questions.

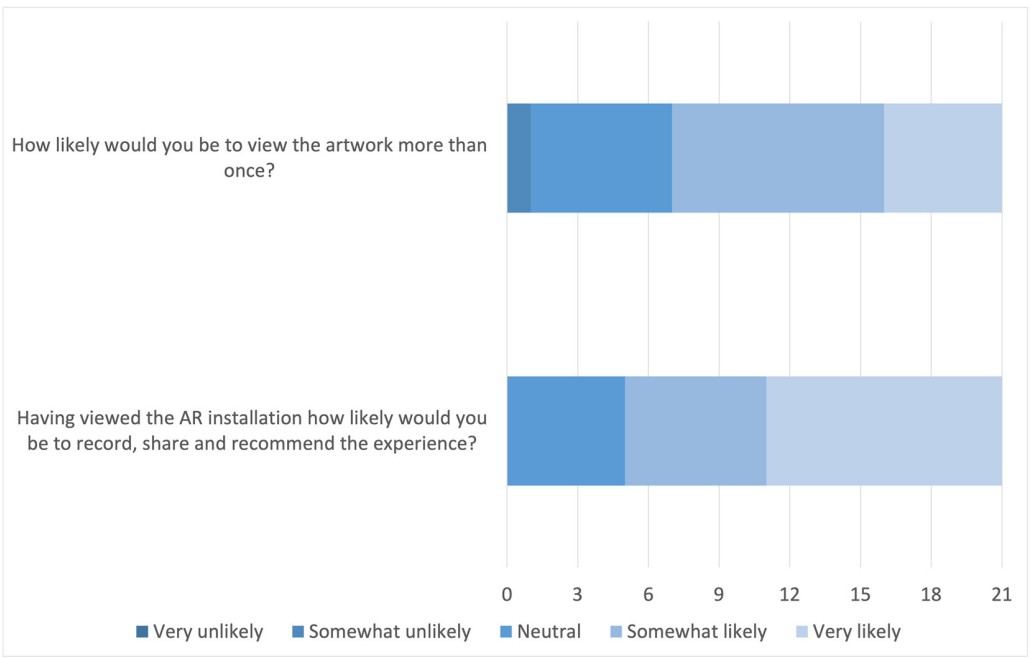

**Figure 7.** Participant responses.

Overall, this study showed us that there is interest in the use of augmented reality art in this way and that users are generally impressed by it. However, we need to be careful in ensuring that it is accessible to users and that they are aware of its presence. Unless and until augmented reality art becomes commonplace, we will have to spend time ensuring that we make people aware of the presence of AR content and how to access it.

## 7. Discussion

All of the work discussed in this paper has focussed on how augmented reality-enabled public art can be used to help people re-engage with derelict urban spaces. There is already an acknowledgment that public art can help re-invigorate abandoned spaces, help restore pride in those spaces and engage the public with the space, encouraging further restoration [30,31]. Through the use of public art, "the unappealing dying spaces of old cities will be transformed into energetic and vibrant little pockets of lively spaces" [32]. Interest in public art also has the possibility to improve local, national and international tourism [33].

### 7.1. Engaging Visitors with Spaces Using Public AR Artworks

Adding an augmented reality layer to public artwork can extend these possibilities further. While at its most basic, AR can add a level of novelty that may increase interest, it can also be used to tell stories in a way that complements and extends the physical artwork. Our work has shown the potential for telling the stories of abandoned places through the combination of AR and physical artworks. As noted by visitors to the Re:Imagined exhibition, the AR layer increased connection with the space. By carefully designing the AR content, we can help engage visitors with the space, with its stories and legends, with its past and present, and perhaps with its future.

To enable this, we recommend that AR-enabled public artworks be designed specifically to consider the physical, social and historical aspects of the sites in which they are to be placed. Simply adding a mural to a site and layering augmented reality content over this may result in an interesting artwork, but if it does not engage with the site itself, then there is a question as to the value of placing it in that space rather than in a gallery. Urban art is often used as part of a regeneration strategy, but this art should engage with the space and the people who reside there, reflect the character of the space and help to combat the depersonalisation that has occurred in many urban spaces [34].

Such artworks can (and possibly should) be collaborative in nature, involving stakeholders that could include residents, local authorities and artists. By including this range of stakeholders, we can create pieces that more thoroughly reflect the space and its people. There may be local history or legends about the space that could be told through the artwork, but without involving the locals, how can we know this? How can we tell the stories of the place without first knowing them?

As well as the artistic implications of the space, we should be aware of the social and physical aspects. Is the space where the artwork will be placed one that is easily accessible? Inaccessible places might have interesting stories, or provide an intriguing experience, but can restrict access to the art. Spaces can have physical or social barriers that stop people from accessing them, from opening times and security guards to "defensive architecture" [35]. Spaces can also be inaccessible to those with mobility issues. Or, spaces can be physically public but have other barriers that make them not truly public spaces in terms of access to information, resources or discussion [36]. For truly public artwork, we need to consider all of these aspects of the site and decide whether the site is accessible enough or whether the location is so significant that it overrides any lack of accessibility.

On top of access to the space, there are considerations regarding the public's ability to move around within the space to inspect the artwork. Is the space so busy that the view is always blocked? Can viewers access the artwork from different angles, something that is particularly important for 3D augmented reality? One of the issues noted with the artworks created in cooperation with Limerick City Council was that views were often obstructed by passing pedestrians and cars. While this project could not be relocated, issues such as these may require a reconsideration of a physical site, particularly for animated artworks that require longer viewing times.

### 7.2. User Experience of Public AR Artworks

Currently, the technology used to display augmented reality art requires the use of smartphones and special apps. While smartphones have become ubiquitous, AR technology is not available by default in them, so sites containing AR artworks will need some means of notifying visitors of the availability of the AR content and how to access it. This could require informational posters, which can detract from the art or a form of organised advertising campaign. Artists creating such works might need to engage with local tourist offices or with groups running walking tours. Some cities already have street art walking tours that could perhaps also show the AR aspect of the artworks. There is also some burden in having to download and install apps, particularly as different artists may work with different apps. Ideally, AR capabilities would, over time, become directly embedded in the workings of smartphones so that the use of the phone camera could notify about and/or activate the augmented reality.

While such integration of augmented reality into our standard interaction with technology may still be some time away, the current state of the technology still allows us to create artworks that can encourage users to engage with the spaces in which they are situated and to transform abandoned spaces into once again lively, active ones—as some of our visitors said at the Re:Imagined exhibition: to "bring them to life".

We suggest that artists should consider how the phone-based interaction affects the experience for viewers. We noted through observation that once visitors are aware of the AR aspect of the artwork, they mostly engage with the art through the lens of the phone. This can disconnect them from the physical artwork. A more meaningful interaction may occur if the visitors are not aware of the augmented reality content until they have looked at the physical artwork or where the augmented reality content adds to the physical artwork in such a way that the two have to be viewed in sequence. However, designing the artwork to work like this can be tricky, and artists may wish to consider if there is a non-phone-based way of augmenting the artwork, such as projected AR, as sometimes used in areas such as the performing arts and cultural heritage [37–39].

Our work has highlighted that many of the barriers to engaging with augmented reality art come from the use of smartphones as the main technology for displaying and interacting with AR content. The interaction of looking through a small screen that the user must hold in front of their face is far from ideal. Feedback from users interacting with our artworks suggests that removing the phone from the interaction would improve the overall experience and that such artworks would benefit particularly from the development of small, lightweight augmented reality headsets. Such headsets would allow for a much more natural interaction with the artwork and a more seamless integration of the physical and AR content.

### 7.3. Tools and Techniques for Creating Public AR Artworks

There are a large number of existing tools for the creation of augmented reality experiences, which can largely be divided into those designed for programmers and those designed for non-programmers [40,41]. Those designed for non-programmers (a group that likely includes the majority of artists interested in the creation of AR artworks) are often general-purpose and offer somewhat basic features, lacking the depth required to allow artists to express their ideas fully. Most of the works discussed in this paper have made use of 2D animations and (sometimes) sound, along with visual markers for triggering content, as these were the capabilities available in current off-the-shelf AR tools.

To enable artists to make use of the potential of AR and place-based storytelling more fully, we require tools that offer more flexibility but that do so in a way that can be integrated into an artist's existing practice and that work with existing tools and workflows. There is much scope here for further research into the features these tools should provide, both in terms of content creation and content distribution. As already discussed, our users raised the need to install apps to view the content as an issue, one that becomes worse if every piece or collection of AR artwork has its own app.

Alongside this, there is a lack of information on the practice of creating augmented reality artworks. While artists are using these technologies, there does not seem to be much discussion of how they are doing so. In this paper, we discuss some aspects of a developing practice so that both artists and AR tool developers can be aware of it and integrate it either into their own practice or into the development of new tools. Understanding how artists are using AR authoring tools can aid in the creation of more useful future tools.

### 7.4. Creating AR-Enable Artworks for Public Spaces

As discussed earlier, central to the creation process for AR public art is the relationship between physical space, still image (illustration) and the AR experience. Here, we discuss our process for developing such artworks, which has been informed by the research presented in this paper. The process can be roughly separated into three stages: concept development, asset development and AR development. Throughout each phase, the illustration and the AR artwork are developed concurrently.

To describe this process, reference is made to a case study: the design of a prototype installation for the outer hoarding wall of an abandoned police station in Mary Street, Limerick. The piece, titled "Outer Inner Space", was inspired by the walking tour of derelict Limerick, during which participants discussed alternative uses for the abandoned building, which once performed a much-needed service of securing the area in this historic part of the city. The concept for the illustration was based on a child's suggestion that the space be transformed "from a police station to a playground, from boring to fun".

During the concept development phase, this idea was sketched and evolved into a large-scale depiction of a kneeling child painting a blue sky onto a section of the hoarding. A primary consideration during this phase was the composition and placement of the physical illustration relative both to the architecture of the site and to the animation that would load as part of the AR experience. In the AR environment, the character would sit up to "paint" a scene from outer space onto the windows on the upper level of the building. Thus, the physical illustration needed to possess both an intrinsic appeal as a standalone site-specific artwork whilst also ensuring that it was placed and scaled to allow the AR animation to play through a natural movement arc within the frame. This is an example of the importance of synergy between the illustration and its AR extension: the former should entice the viewer to pause and engage with the site, offering a moment of the story, which is fully revealed and contextualised through the latter.

Viewer safety was a further consideration in this stage of the design process. The illustration was placed in a manner that the viewer would be able to frame the illustration from the correct angle through their camera while standing on the sidewalk, without having to step into the street.

Whereas the concept development phase is concerned with identifying and solving problems related to the idea and the location. The asset development stage looks at the production of the audio and visual assets that comprise both the Illustration and the AR artwork. The sketched illustration is scanned into the computer to be refined and developed digitally. Given the required scale of the illustration, it is ideally drafted either in a vector-based drawing program or as a large-scale Photoshop document. An important consideration during this stage of the process is to develop the various elements in a way that will allow them to be animated individually or to stack on top of each other, creating a sense of depth in the AR experience.

Once the illustration is finalised, it is saved as a flattened, scaled-down.jpg file suitable for animation in Adobe After Effects or the Photoshop timeline. The illustration essentially becomes the marker image and the first frame of the animated story. In addition to the animation, the voiceover/audio is recorded, and any still images are exported as PNGs with a transparent background. A challenge here is to design the animation in a way that obscures the physical illustration when it loads over the marker in the AR app. For example, in this case study, the crouching figure of the child character featured in the illustration would still be visible beneath the animation playing in the AR application. Thus, an object

layer, a solid color or an image of the blank wall, would be introduced into the background of the animation to obscure the physical illustration of the child when the animation loads.

These assets, including the marker image, are then imported into the AR app for the final stage of the design process. The AR development phase involves layering the assets in the AR build space, aligning them with the marker image and adjusting their scale and alignment in Z space. This final stage of the process involves frequent user tests ensuring that the marker image is clearly recognisable, that the story unfolds seamlessly for the viewer and that the message is clearly communicated through the relationship between the illustration and its AR extension.

## 8. Conclusions

The work discussed in this paper is part of the ongoing development of an augmented reality art practice that seeks to use AR to tell the stories of places and encourage the public to engage with the places and their stories. In this paper, we described several aspects of this work in regard to working with augmented reality, the use of public artworks and murals as triggers for AR content, walking as an ethnographic and creative practice in the creation of situated AR artworks, and the user experience of such artworks. We believe these can be useful in the design and implementation of future AR artworks in public spaces and in the creation of AR authoring tools.

From this work, we have created some guidelines for both artists interested in augmenting physical, place-based art and for developers working on technologies to support augmented reality. For artists, we recommend the following: (1) integrate the artwork with the space, taking into account the physical and social aspects of the public space; (2) find ways to make visitors or passers-by aware of the augmented reality content that augments the physical artwork without distracting from it; and (3) ensure that the story being told by the augmented reality adds to the story told by the physical artwork and the space in which it is presented. Following these guidelines should help artists develop immersive, engaging augmented reality artworks that integrate with the space in which they are being shown.

With regards to those developing technology to support artists in using augmented reality, particularly in public spaces, we recommend the following: (1) integrate augmented reality with existing tools and workflows that artists are familiar with to allow artists to engage with AR more fully; (2) consider how they can lower the barrier to viewers accessing AR content, whether through integrating AR more fully into existing devices such as smartphones, or creating new, non-smartphone devices that present AR in a more transparent way.

We believe these guidelines can be useful in the design and implementation of future AR artworks in public spaces and also in the creation of AR authoring tools and display technologies. We will continue to develop the practice described here, and future works will include the development of larger-scale AR place-based storytelling works.

**Supplementary Materials:** The following supporting information can be downloaded at: https://www.mdpi.com/article/10.3390/mti7090089/s1, Video S1: Oblivious—AR recording; Video S2: SAP Drawing and tech AR sketchbooks; Video S3: Waterford Walls; Video S4: Reimagine_Mary Street recording.

**Author Contributions:** Conceptualisation, T.Y. and M.T.M.; methodology, T.Y. and M.T.M.; investigation, T.Y.; data curation, T.Y. and M.T.M.; writing—original draft preparation, M.T.M.; writing—review and editing, T.Y.; supervision, M.T.M. All authors have read and agreed to the published version of the manuscript.

**Funding:** This research received no external funding.

**Institutional Review Board Statement:** The study was conducted in accordance with the Declaration of Helsinki and approved by the Ethics Committee of the University of Limerick (2 May 2022).

**Informed Consent Statement:** Informed consent was obtained from all subjects involved in the study.

**Data Availability Statement:** The data presented in this study are available on request from the corresponding author. The data are not publicly available due to privacy reasons. The audio recordings were deleted after transcription for privacy reasons.

**Acknowledgments:** We would like to acknowledge the technical staff of LSAD and UL for their help in printing and installing the exhibition at the respective venues.

**Conflicts of Interest:** The authors declare no conflict of interest.

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
