# Peer review of "An Investigation of the Use of Augmented Reality in Public Art"

_mti, doi:10.3390/mti7090089_

Round 1

Reviewer 1 Report

This paper deals with Augmented Reality applied to public art, trying to provide some guidelines for a wider spread of this technology in such a context.

I provide my comments below:

1. The structure of the paper is unconventional, there is a strong need to describe the content of each Section in the Introduction.

2. The usage of Augmented Reality in Cultural Heritage is spreading more and more. Recently, some analysis on existing solutions have been provided and could be mentioned to better frame the context. As an example: Innocente et al., "A framework study on the use of immersive XR technologies in the cultural heritage domain".

3. When writing about the scope of the paper, the author mentioned a "design approach". I did not find a description of a general "design approach", but only some considerations made for each of the events (workshops, walks, and public art collaborations).

4. Some considerations about the three categories to track the digital content with regards to the real world: the marker-based approach makes use of 2D markers, but could also employ 3D markers to further improve the tracking capabilities. This is particularly important not only to track objects from frames, but also to compute physical properties such as translations and rotations, as explained, for instance, from Marullo et al., "6D object position estimation from 2D images: a literature review".

5. There is a lack of information regarding workshops and walks: how many participants for each workshop/walk? How did the authors gather the data that led to their considerations?

6. How did the authors gather feedback regarding public art collaborations?

7. The summary should be expanded collecting all the common traits found in the previous subsections and should be discussed in the next Section.

8. Explain which questions have been proposed to assess "Re:Imagined".

9. A discussion of AR using Head-Mounting Displays (HMDs) should be included. AR cannot be defined including only smartphones and I think that art could benefit even more from HMDs usage.

English language is fine, just a few typos to check.

Author Response

We would like to thank the reviewer for their very useful comments and feedback. We will address them individually below. All new content in the paper has been highlighted in red.

1. The structure of the paper is unconventional, there is a strong need to describe the content of each Section in the Introduction.

We have changed the titles on the sections and subsections to be more conventional and added a description of each section in the introduction. 

2. The usage of Augmented Reality in Cultural Heritage is spreading more and more. Recently, some analysis on existing solutions have been provided and could be mentioned to better frame the context. As an example: Innocente et al., "A framework study on the use of immersive XR technologies in the cultural heritage domain".

Our focus is more on public art and AR artworks rather than cultural heritage. We have added some additional context to the introduction section to better explain this.

3. When writing about the scope of the paper, the author mentioned a "design approach". I did not find a description of a general "design approach", but only some considerations made for each of the events (workshops, walks, and public art collaborations).

We have removed the term design approach and replaced it with "process". You are correct that we are not presenting a general approach, but rather the process we have taken in our work. We hope this is clarified by the change.

4. Some considerations about the three categories to track the digital content with regards to the real world: the marker-based approach makes use of 2D markers, but could also employ 3D markers to further improve the tracking capabilities. This is particularly important not only to track objects from frames, but also to compute physical properties such as translations and rotations, as explained, for instance, from Marullo et al., "6D object position estimation from 2D images: a literature review".

We have added some text and a reference on 3D markers and an explanation of our choice of 2D image-based markers.

5. There is a lack of information regarding workshops and walks: how many participants for each workshop/walk? How did the authors gather the data that led to their considerations?

We have added data on the participants and how we gathered their feedback in the relevant sections.

6. How did the authors gather feedback regarding public art collaborations?

Feedback was through conversations with organisers, artists and sponsors. We have added text to reflect this.

7. The summary should be expanded collecting all the common traits found in the previous subsections and should be discussed in the next Section.

We have removed the summary heading in Section 4 as it was unnecessary and the findings are included in the discussion section.

8. Explain which questions have been proposed to assess "Re:Imagined".

We have added a discussion of our research questions at the start of Section 5.

9. A discussion of AR using Head-Mounting Displays (HMDs) should be included. AR cannot be defined including only smartphones and I think that art could benefit even more from HMDs usage.

We strongly agree that AR art would benefit from the use of HMDs. We have added a paragraph in section 6.2 to reflect this.

Reviewer 2 Report

Dear authors,

The paper is structured correctly and the argumentation is coherent. The topic is interesting and the illustrations used are relevant. I believe that the work may be considered for publication, with further comments added to the Discussion on the issue of matching the original artwork with the augmented reality intervention.

Author Response

We thank the reviewer for their review. We have added a section (6.4) discussing our process for matching the AR and physical artworks as suggested.

Reviewer 3 Report

1.      A brief summary: The paper presents research on the use of augmented reality (AR) to expand the image-making process. It uses AR in a practical way in art.

2.      General concept comments:

a)     Weaknesses of the paper: more descriptive character of the paper

b)     Strengths of the paper: practical idea – providing a new solution using augmented reality  

c)      Hypotheses / goals / research gap: the research questions are mentioned but they should be clarified, I suggest placing them at the end of the Introduction or in the Methodology section

d)     Methodology: no separated section – it would be worth adding it

e)     Literature: 39 references – the literature is well-selected and appropriate   

3.      Specific comments:

In Figure 7, it is worth naming the axes.

I suggest that all subheadings in chapters should also be numbered according to the template.

The article is more descriptive; it would be justified to organize it better – the research part sections and subsections should be clearer and more logically organized. It would be easier for the reader to understand the achievement of the research.

In the abstract, the abbreviations in parentheses (AR) should appear after the first appearance of "augmented reality".

In the Introduction, when a quote appears, it should also be explained, not just appear.

The text of the introduction should be aligned.

The paper is worth publishing – after some improvements.

It should be emphasized that additional materials in the form of videos are worth seeing because they show in a very realistic way the subject of research and the novelty of the application of augmented reality in art.

Author Response

We thank the reviewer for their useful comments. We are particularly pleased that they liked the additional video materials. We have made changes to the paper to reflect the comments. Changes appear in red text. In response to specific comments:

the research questions are mentioned but they should be clarified, I suggest placing them at the end of the Introduction or in the Methodology section

We have added some clarification of the research questions in the introduction and also in Section 5 where we include the specific research questions for the exhibitions.

Methodology: no separated section – it would be worth adding it

I suggest that all subheadings in chapters should also be numbered according to the template.

The article is more descriptive; it would be justified to organize it better – the research part sections and subsections should be clearer and more logically organized. It would be easier for the reader to understand the achievement of the research.

We have changed the section headings to be more standardized and include a methods section, added section numbers, and included a description of each section in the introduction.

In the abstract, the abbreviations in parentheses (AR) should appear after the first appearance of "augmented reality".

We have fixed this as suggested

In the Introduction, when a quote appears, it should also be explained, not just appear.

We have included the quote in the text of the introduction so that it is explained in context.

Round 2

Reviewer 1 Report

I very much appreciated the effort spent in addressing my doubts. In my opinion the manuscript can now be accepted in the present form.

Reviewer 3 Report

Thank you for improving the paper. I accept the changes.